# A Scientific Approach for Environmental Analysis: An Asynchronous Electric Motor Case Study for Stand-By Applications

Andrea Antonacci [ID], Alessandro Giraldi [ID], Eleonora Innocenti and Massimo Delogu *[ID]

Department of Industrial Engineering, University of Florence, Via di S. Marta 3, 50139 Florence, Italy; andrea.antonacci@unifi.it (A.A.); alessandro.giraldi@unifi.it (A.G.); eleonora.innocenti@unifi.it (E.I.)
* Correspondence: massimo.delogu@unifi.it; Tel.: +39-055-275-8769

**Abstract:** In recent years, there has been growing attention from the scientific community regarding the environmental impact of commercial goods, pushing companies to adopt life cycle assessment strategies to improve their environmental profile. Only few studies have examined the environmental burdens of electric motors, specifically for stationary applications such as oil and gas plants, transmission organs, operating machines, or other industrial utilization. For this purpose, this paper presents a comprehensive and detailed evaluation of the environmental sustainability of an asynchronous electric motor used for stationary applications. The motor under examination moves a stand-by hydraulic pump used in a compression plant to lubricate the bearings of centrifugal machines. The principles dictated by ISO 14040 are implemented, and a cradle-to-gate analysis is performed. This article reports in detail the inventory hypothesis and the steps that led to modeling the assessed electric motor. The results are presented for all impact categories provided by the ReCiPe methodology. Additionally, a breakdown of the eco-indicators at the single component level is proposed, focusing on the impact of raw material extraction phases and subsequent technological processes. The last section highlights which components contribute predominantly, both from a materials and processes perspective, and the environmental hotspots in the modeled supply chain are identified.

**Keywords:** electric motor; oil and gas; stand-by applications; life cycle assessment (LCA); mechanical design; industrial engineering; sustainability





## 1. Introduction

In the current industrial scenario, increasing attention is being paid to the environmental sustainability aspects of commercial products [1]. This trend is due to government regulations [2] aimed at reducing emissions, and market strategies, where product sustainability is increasingly becoming a qualifier for companies [3]. In this context, there is a strong interest in the electric motor sector, which consumes approximately 40% of the electricity produced worldwide, contributing approximately 13% of the total $CO_2$ generated [4]. Among the impact phases of a product's life cycle, the manufacturing phase makes a not-negligible contribution [5]. Therefore, it is crucial for manufacturers to ensure that this product category follows the principles of integrating eco-design aspects [6].

In this regard, dedicated tools are available to help evaluate the environmental sustainability of the designed electric motor with a high-level approach [7]. The literature focuses heavily on the study of electric traction motors [8–11], given the growing expansion of electric mobility [12]. Among them, a very in-depth study is devoted to the production phase of permanent magnet motors using materials such as ferrites, NdFeB alloys, and so on [13,14]. Nordelöf et al. [15] in their work conducted a comparative life cycle assessment study for three different permanent synchronous magnet traction motors: one with neodymium–dysprosium magnets, one with samarium–cobalt magnets, and one with magnets made of ferrite alloy. The study demonstrates that production is the main contributor

to the environmental impact per kilometer, both in terms of $CO_2$ and human toxicity, in the Swedish geographic scenario. Contrariwise, the usage phase has a higher percentage impact in the American scenario. The impact of the production phase mainly depends on the contributions of aluminum and copper for frame production rather than the material required to produce the magnets. Del Pero et al. [16] conducted a *cradle-to-grave* study on a permanent magnet motor used in motorsports, assessing its environmental impact in terms of global warming potential (GWP) and primary energy demand (PED). Once again, the study shows that production is the main contributor to environmental impact, with around 70% coming from raw materials. The analysis provides a detailed breakdown of the impacts on the various components of the motor, revealing that over 70% of the production impact is associated with the production of the stator components. Tintelecan et al. [17] developed a comparative environmental assessment study of a synchronous motor and a permanent magnet motor using the Impact Assessment ReCiPe 2016 method. With both motors having the same external dimensions, the synchronous motor has a higher impact on all evaluated categories than the permanent magnet motor. For both configurations, aluminum components mainly contribute to the environmental impact. Indeed, several studies focus on the environmental profile contingent on the efficiency of the usage phase of the motor. Orlova et al. [18] investigated the dependence of environmental impacts during the usage phase on the efficiency of three traction motors. The permanent magnet motor under study was found to be the most impactful in terms of production, but this is compensated for by the higher efficiency during the usage phase of the other two motors considered. There is also the study by Autsou et al. [19], which deals with implementing a digital twin in the Simulink environment for the sustainability assessment of induction or synchronous reluctance electric motors. This model aims to evaluate efficiency for future life cycle assessment (LCA) evaluations.

Regarding stationary motors, several articles study aspects of design [20–22]. Fewer articles focus on their environmental profile. Even in this case, studies can be divided based on the assessment of the production phase or the environmental impact of the usage phase in terms of energy efficiency [23]. Boughanmi et al. [24] reported an LCA study of an asynchronous electric motor with a squirrel cage design, detailing 10 different impact categories. The study analyzes two different usage scenarios for the motor: the first scenario assumes a usage of 1000 h, and in this case, production is found to be the most impactful phase from an environmental point of view. In the second scenario, the duration analyzed is 20,000 h; in this case, the usage phase is dominant compared to the production phase. In the work of Auer et al. [25], the environmental profile of three stationary electric motors is extensively studied according to ISO14040-14044 [26,27] standards. The motors under study have a nominal power of 110 kW, are made of cast iron, and have four poles. The usage phase is evaluated for 20 years and is found to be the most impactful phase for all the analyzed motors. The study demonstrates that an increase in efficiency leads to benefits in all the studied impact categories. Indeed, some studies delve into the end-of-life phase of electric motors, such as the work by Jerome et al. [28], where the benefits of life extension are highlighted in terms of GWP and resource depletion obtained from the Ecoinvent 3.3 database.

That said, this study analyzes the environmental profile of a squirrel cage asynchronous motor used in an industrial application to power a stand-by hydraulic circuit. The motor under study has a power output of 30 kW to operate a centrifugal pump used in an oil and gas plant. The analysis is supported by the implementation of a three-dimensional model of the motor based on primary inventory data collection. The motor profile is analyzed in detail for each component and its respective manufacturing process. The results are presented for all 18 impact categories provided by the ReCiPe 1.13 impact assessment methodology to avoid shifting impacts to other categories. The Ecoinvent v.3.9 [29] database is used for the study.

Therefore, in light of the above state-of-the-art review, this paper aims to implement the following crucial aspects:

- *Modeling tree inventory data implementation*: implementing a top-down scientific approach for quantitative modeling, in order to overcome the lack of sensitive data for industrial applications.
- *Cradle-to-gate analysis*: focusing on detailed manufacturing stage modeling and highlighting results that are not noticeable when a cradle-to-grave evaluation is performed.
- *Comprehensive sustainability assessment*: choosing a ReCiPe impact assessment with 18 impact categories and not simply assessing traditional categories like GWP.
- *Sustainability hotspots*: providing suggestions for manufacturers and clients to reduce environmental burdens.
- *Guidelines for future parametrized analysis*: proposing detailed evaluations of electric motor classification from a sustainable point of view, to be used as a basis for a comprehensive scalable assessment.

The sections are structured as follows: in the Materials and Methods, we describe the approach to modeling for inventory development and its impact assessment, and then the obtained results are discussed in order to highlight the main conclusions.

## 2. Materials and Methods

The sustainability assessment follows the LCA methodology, supported by and ISO14044 standards. This framework allows the evaluation of the sustainability profile of a product throughout its entire life cycle. Therefore, this evaluation is assessed by quantifying the material and energy flows that the system exchanges with the external environment throughout its life. The methodology involves defining four distinct points, described as follows (reported in Figure 1).

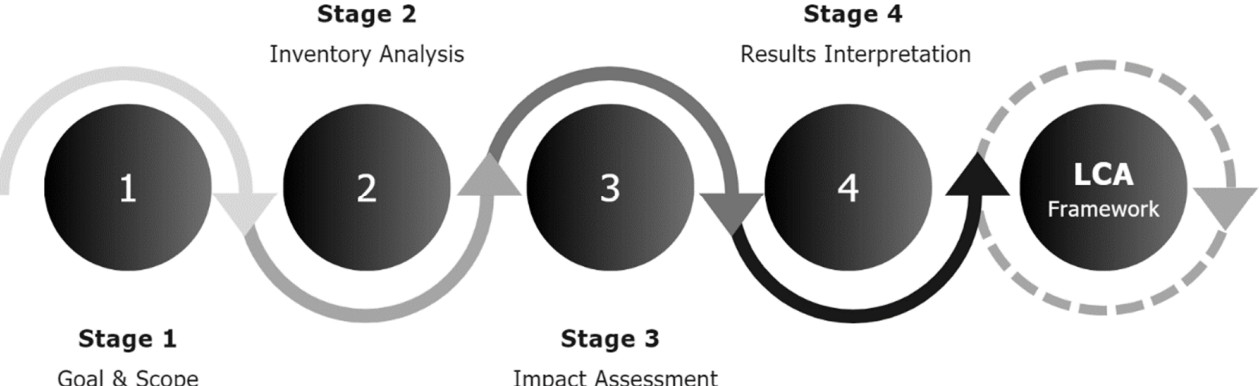

**Figure 1.** LCA framework and interaction between phases of the study.

**Goal and Scope.** The objective of this analysis is to evaluate the environmental sustainability of an asynchronous electric motor (AEM) with a squirrel cage design. This motor is used to power a pump in an oil circuit for lubricating the bearing pads of a centrifugal compressor. In this context, the functional unit used in this study is the production of AEM with 30 kW of output rotary power to supply the stand-by pump at the required speed.

The environmental profile of the motor is assessed using the ReCiPe Midpoint (H) 1.13 impact assessment methodology [30] in terms of the following 18 impact categories (Figure 2):

A brief description of these categories is provided in the Supplementary Materials. In order to give results that are easier to verify and report, the midpoint indicators are used [31]. Indeed, in contrast with endpoint indicators, these assessments do not introduce further arbitrary hypotheses and are suitable for identifying environmental hotspots, one of the main goals of this paper. In this regard, the ReCiPe methodology is chosen between all the other approaches available, like CML2002, LIME, or TRACI, due to the considerable use of the LCA analysis of industrial products, the recent update in 2016 and the broader list of midpoint indicators included [30].

**Figure 2.** ReCiPe midpoint (H) categories.

The system boundaries analysed are encompassed between the *cradle-to-gate* stages, described in Table 1. Therefore, the analysis takes into account the contributions from raw material extraction (RAW) and manufacturing and assembly processes (MAN), including transportation along the specific supply chain. This inclusion is achieved by utilizing data from the "market" category of the Ecoinvent 3.9 database, where the processes within the category encompass both the transport phase and other relevant aspects.

**Table 1.** System boundaries of cradle-to-gate analysis.

| System Boundaries | | |
|---|---|---|
| **LC Stage** | **LC Sub-Stage** | **Processes** |
| Cradle-To-Gate Analysis | Raw Materials (RAW) | • Production of electricity, heat, steam and fuel for raw material extraction and production.<br>• Raw material extraction and primary production processes.<br>• Fuel/energy production for the transportation of raw materials between suppliers' plants. |
| | Manufacturing (MAN) | • Production of electricity, heat and auxiliary material for manufacturing and assembly activities.<br>• Manufacturing and assembly processes.<br>• Fuel/energy production for the transportation of components between suppliers' plants.<br>• Recovery processes of scrap materials from manufacturing activities. |

**Inventory Analysis and Impact Assessment Modeling.** The implementation of inventory data for the AEM and its parts is derived from the modeling process of the selected component by submitting a specific technique. The process is well described by the modeling tree (presented in Figure 3), whose steps are outlined below.

- **Primary E-motor Critical-To-Quality (CTQ) data.** The primary available data pertain to the characteristics of the electric motor, which must power a hydraulic pump with a power output of 25 kW. Additionally, the dimensions of the motor required to ensure its correct positioning are also available. The motor is of a squirrel cage design, with a three-phase asynchronous configuration. It produces a high-efficiency (IE2) power output of 30 kW, with a rotation speed of approximately 3600 rpm and F insulation class. The motor dimensions measure $402 \times 842 \times 400$, with a total weight of approximately 304 kg. These specifications summarized in Table 2 are crucial factors

in determining the motor's suitability and quality for the intended application, and they form the basis of the inventory data necessary for its modeling.

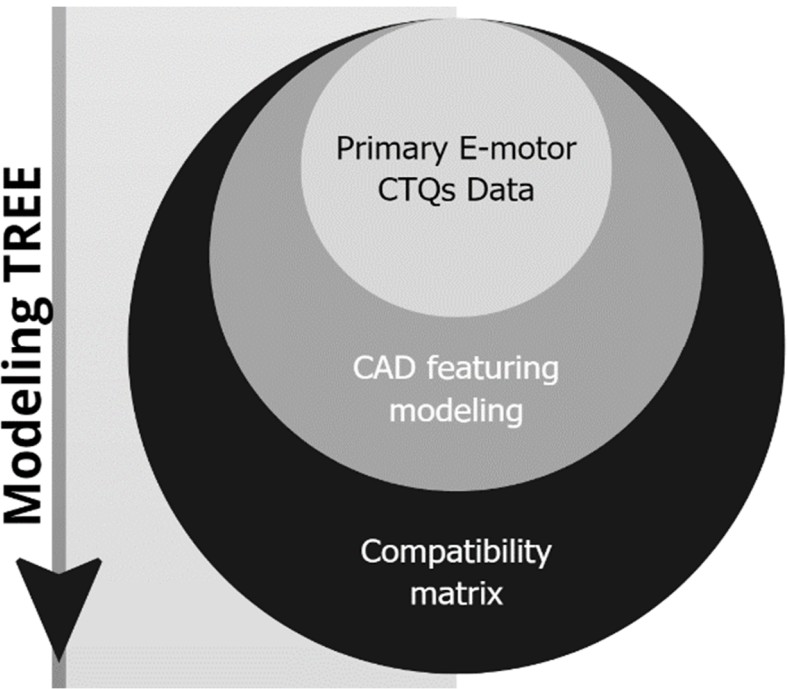

**Figure 3.** Modeling tree of inventory data implementation.

**Table 2.** Electric motor specifications. The full name plate of the electric motor cannot be included due to an active non-disclosure agreement.

| Specifications | Value | Unit |
|---|---|---|
| Power | 30 | kW |
| Rotational Speed | 3600 | rpm |
| Pole | 2 | - |
| Nominal voltage | 400 | V |
| Frequency | 60 | Hz |
| Full-Load Current | 54 | A |
| Full-Load Torque | 97 | Nm |
| Efficiency Class | IE2 | - |
| Insulation Class | F | - |
| Dimensions | $402 \times 842 \times 400$ | mm |
| Weight | 304 | kg |

- **CAD feature modeling.** Based on the known data pertaining to the motor parameters, a three-dimensional modeling process was carried out to obtain inventory data using the assembly's geometric features, since, according to the top-down scientific approach, it is crucial to overcome the possible lack of primary data of the analyzed system. This modeling process involves the integration of various data inputs, including the motor's design specifications, dimensions, and material composition, to create a comprehensive virtual 3D representation of the motor and its components. Using advanced software tools and techniques, the modeling process can accurately capture the motor's functional and structural characteristics, allowing for a detailed analysis of its environmental impact throughout its life cycle. The main motor components

are shown in Figure 4 with an exploded view in the CAD environment; the presented assembly is divided into sub-assemblies in Table 3, which group together components with similar functions. The sub-assembly grouping provides a clear overview of the motor's components and their functions, facilitating a more detailed and comprehensive analysis; thus, by breaking down the motor into smaller sub-assemblies, it is possible to focus on specific components and their associated environmental impacts. Due to the difficulties in representing results and considerations with the full component names, a list of the simplest acronyms is shown in Table 3. More intuitive abbreviations will be implemented in the next paper, considering a combination of component, sub-assembly and material composition [32].

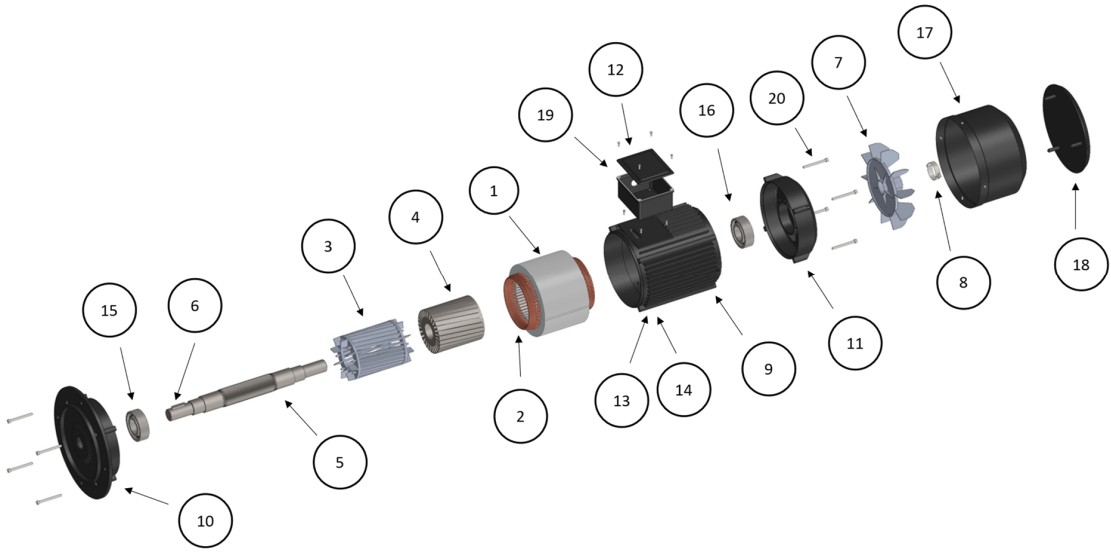

**Figure 4.** AEM CAD—exploded view.

**Table 3.** Breaking down of electric motor assembly in peculiar sub-assemblies.

| Assembly | Sub-Assembly | Components | Acronym | ID | Mass (%) |
|---|---|---|---|---|---|
| AEM | STATOR | Stator Core Laminations | SCL | 1 | 23.62% |
| | | Wirings—Filaments | WIR | 2 | 4.94% |
| | | Wirings—Insulation | | | 0.39% |
| | ROTOR | Rotor Squirrel Cage | RQC | 3 | 2.11% |
| | | Rotor Core Laminations | RCL | 4 | 10.11% |
| | | Rotor Shaft | RSH | 5 | 7.38% |
| | | Key Shaft | KSH | 6 | 0.05% |
| | | Fan | FAN | 7 | 2.04% |
| | | Fan Clamps | FAN CL | 8 | 0.03% |
| | FRAME | Electric Motor Case | EMC | 9 | 15.63% |
| | | Flange Drive-End Shield | FDS | 10 | 11.53% |
| | | Non-Drive-End Shield | NDE E | 11 | 8.05% |
| | | Terminal Box | TB | 12 | 2.15% |
| | | Grease Fitting | GF | 13 | 0.01% |
| | | Grease Fitting Protection | GFP | 14 | 0.00% |
| | | Bearing Drive-End Shield | BDS | 15 | 0.85% |
| | | Bearing Non-Drive-End Shield | B NDE E | 16 | 0.85% |
| | | Fan Cover | FAN CO | 17 | 5.97% |
| | | Drip Cover | DRIP CO | 18 | 3.19% |
| | MIX | Cables | CAB | 19 | 0.80% |
| | | Miscellaneous (Gaskets, Screws, etc.) | MISC | 20 | 0.29% |

Figure 5 illustrates the percentage distribution of the AEM mass among the previously defined sub-assemblies, emphasizing that the FRAME constitutes the most significant sub-assembly compared to others, accounting for approximately 48.5% of the total mass. The AEM masses of each component are presented in Table 3.

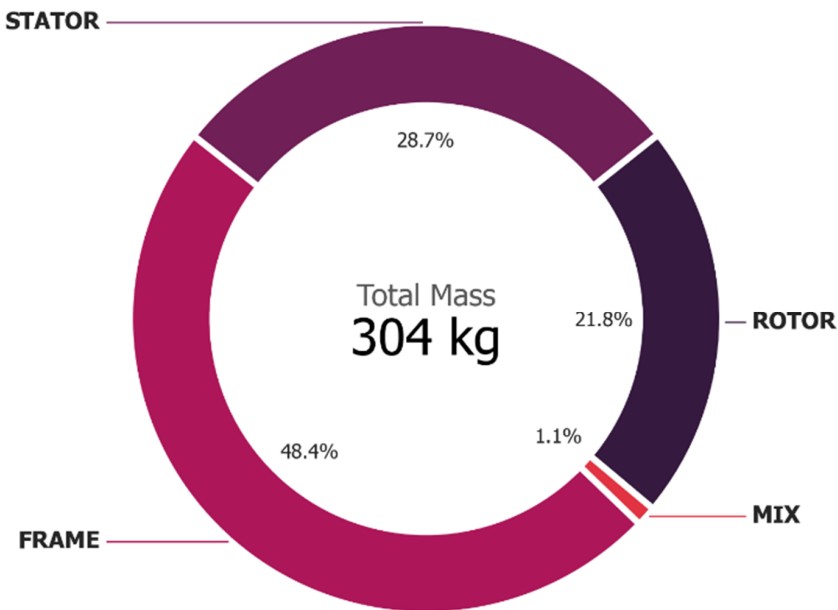

**Figure 5.** Mass breakdown of AEM assembly in specific sub-assemblies.

**Compatibility Matrix.** Based on the breakdown of the assembly, Table 4 presents the association of each component with the information necessary to implement the *cradle-to-gate* analysis. The information pertains to the production processes that lead to the generation of the component and is based on the constraints imposed by the material–process compatibility matrices [33]. These matrices ensure that the manufacturing processes used for each component are compatible with the material composition.

**Table 4.** Association of materials and main manufacturing processes with AEM components.

| Sub-Assembly/Components | | Cradle-to-Gate Approach | | | |
|---|---|---|---|---|---|
| | | Material Stage | Manufacturing Process Stage | | |
| | | | Primary Process | Secondary Process | Surface Treatment |
| STATOR | Stator Core Laminations | Steel, low-alloyed | Hot Rolling, Cold Rolling, $CO_2$ Laser | - | - |
| | Wirings—Filaments | Copper | Wire Drawing, Bending | - | - |
| | Wirings—Insulation | Epoxy Resin, Plastic Film | - | - | - |
| ROTOR | Rotor Squirrel Cage | Aluminum, Primary, Ingot | Die Casting | - | - |
| | Rotor Core Laminations | Steel, Low-Alloyed | Hot Rolling, Cold Rolling, $CO_2$ Laser | - | - |
| | Rotor Shaft | Steel, Low-Alloyed | Hot Rolling | Turning | - |
| | Key Shaft | Steel, Low-Alloyed | Hot Rolling, $CO_2$ Laser | - | - |
| | Fan | Cast Iron | Sand Casting | Turning, Drilling | - |
| | Fan Clamps | Steel, Low-Alloyed | Hot Rolling, Cold Rolling | Drilling | - |

**Table 4.** *Cont.*

| Sub-Assembly/Components | | Cradle-to-Gate Approach | | | |
|---|---|---|---|---|---|
| | | Material Stage | Manufacturing Process Stage | | |
| | | | Primary Process | Secondary Process | Surface Treatment |
| FRAME | Electric Motor Case | Cast Iron | Sand Casting | Milling, Drilling | Primer Painting, Epoxy Painting, Finishing Painting |
| | Flange Drive-End Shield | Cast Iron | Sand Casting | Milling, Drilling | Primer Painting, Epoxy Painting, Finishing Painting |
| | Non Drive-End Shield | Cast Iron | Sand Casting | Milling, Drilling | Primer Painting, Epoxy Painting, Finishing Painting |
| | Terminal Box | Cast Iron | Sand Casting | Milling, Drilling | Primer Painting, Epoxy Painting, Finishing Painting |
| | Grease Fitting | Steel, Low-Alloyed | Hot Rolling, Forging | - | - |
| | Grease Fitting Protection | Polyethylene, Low Density | Injection Molding | - | - |
| | Bearing Drive-End Shield | Steel, Low-Alloyed | Hot Rolling, Forging, $CO_2$ Laser | Milling | - |
| | Bearing Non-Drive-End Shield | Steel, Low-Alloyed | Hot Rolling, Forging, $CO_2$ Laser | Milling | - |
| | Fan Cover | Cast Iron | Hot Rolling, Cold Rolling, Deep Drawing | $CO_2$ Laser | Primer Painting, Epoxy Painting, Finishing Painting |
| | Drip Cover | Cast Iron | Hot Rolling, $CO_2$ Laser | - | Primer Painting, Epoxy Painting, Finishing Painting |
| MIX | Cables | Cable Material | Cable Manufacturing | - | - |
| | Miscellaneous (Gaskets, Screws, etc.) | Material Mix | Mix Manufacturing | - | - |

The materials and processes are modeled by associating them with the Ecoinvent v3.9 database, and the corresponding outcomes are presented in specific tables in the Supplementary Materials. This approach ensures that accurate and comprehensive data regarding materials and processes are incorporated, enabling a robust analysis and evaluation of the system's environmental impact.

Regardless of the methodology chosen for the LCIA, the environmental impact (EI) calculation of a generic product (or component) follows the modeling framework described in Table 5 (see Equations (1)–(3)). Such a calculation is derived by considering the comprehensive *cradle-to-gate* LC of the component, encompassing the following main stages:

- Materials: this encompasses the impacts from raw material extraction up to the manufacturing of semi-finished products.
- Manufacturing: this addresses the impacts of the primary manufacturing processes necessary to convert semi-finished products into the final component, as well as the secondary processes and surface treatments.

The presented framework will be used for each AEM component, obtaining the results reported in the following paragraphs.

**Table 5.** *Cradle-to-gate* equations.

| *Cradle-to-Gate* Equations | |
|---|---|
| $EI^{PROD} = EI^{MAT} + EI^{MAN}$ | (1) |

| MATERIAL (MAT) | $EI^{MAT} = m \cdot ei^{MAT}$ | (2) |
|---|---|---|
| MANUFACTURING (MAN) | $EI^{MAN} = \left( \sum\limits_{i}^{N_{PR}} ei_i^{PR} \cdot l_i^{PR} + \sum\limits_{j}^{N_{JO}} ei_j^{SE} \cdot l_j^{SE} + \sum\limits_{k}^{N_{SU}} ei_k^{SU} \cdot l_k^{SU} \right)$ | (3) |

$EI^{PROD}$ = environmental impact of the production phase (impact-category-defined).
$EI^{MAT}$ = global warming potential of the material phase (impact-category-defined).
$EI^{MAN}$ = global warming potential of the manufacturing process phase (impact-category-defined).
m = mass of the component/product (kg)
$ei^{MAT}$ = mass-specific EI in the material stage phase (impact-category-defined/kg)
$ei^{PR}_i$ = process-specific EI in primary processes for specific process i (impact-category-defined/process-defined)
$ei^{SE}_j$ = process-specific EI in secondary processes for specific process j (impact-category-defined/process-defined)
$ei^{SU}_k$ = process-specific EI in surface treatment processes for specific process k (impact-category-defined/process-defined)
$l^{PR}_i$ = characteristic parameter associated with primary processes for specific process i (process-defined)
$l^{SE}_j$ = characteristic parameter associated with secondary processes for specific process j (process-defined)
$l^{SU}_k$ = characteristic parameter associated with surface treatment processes for specific process k (process-defined)
$N_{PR}$ = number of primary processes associated with component/product (-)
$N_{SE}$ = number of secondary processes associated with component/product (-)
$N_{SU}$ = number of surface treatment processes associated with component/product (-)

## 3. Results

The presented outcomes and figures provide valuable insights into the environmental impacts of different electric motor components; Table 6 reports 8 of all 18 indicators defined by the ReCiPe methodology, and the other impact categories can be found in the Supplementary Materials.

Let us initially focus on GWP, the impact category most commonly used in LCA studies, and then on the other categories. Figure 6 illustrates, through a bar chart, the GWP results for each AEM component, clearly stressing the impacts during the raw material (RAW) and manufacturing (MAN) phases. Additionally, the impacts are classified from the most impactful component to the least impactful. It is clear that there is no phase of the life cycle that can be considered inherently more important than another. The results reveal that specific components have high impacts during the RAW phase but relatively lower values during the MAN stage. For instance, the drip cover component (DRIP CO) demonstrates a low impact due to the hot-rolling process during manufacturing, but a higher material impact due to cast iron production. Contrariwise, the motor case component (EMC) depicts a trend reversal, with a higher impact during the manufacturing phase than the raw material phase due to the sand-casting process.

Figure 7 depicts a cumulative GWP and mass representation of the AEM components. These, when summed together, are arranged based on the order obtained from the classification of GWP in Figure 6. In this context, the impact of the first five (out of twenty) components corresponds to almost 65% of the total environmental impact of the motor. Considering the mass perspective, these five elements represent almost 58% of the overall mass. This finding highlights that 20% of the total motor components significantly contribute to the system's overall mass composition and environmental footprint.

**Table 6.** First 8 out of all 18 ReCiPe impact categories outcomes for each AEM components.

| Sub-Assembly/Components | | ReCiPe Midpoint (H) 1.13 | | | | | | | |
|---|---|---|---|---|---|---|---|---|---|
| | | GWP$_{100}$ | FDP | MEP | ALOP | FETP$_{inf}$ | HTP$_{inf}$ | MDP | WDP |
| | | kg CO$_2$-Eq | kg oil-Eq | kg N-Eq | m$^2$-year | kg 1,4-DCB-Eq | kg 1,4-DCB-Eq | kg Fe-Eq | m$^3$ Water-Eq |
| STATOR | Stator Core Laminations | 225.56 | 62.79 | 0.06 | 9.01 | 12.25 | 108.02 | 130.37 | 2.69 |
| | Wirings—Filaments | 108.07 | 31.31 | 0.09 | 16.62 | 343.56 | 2095.46 | 571.31 | 2.23 |
| | Wirings—Insulation | 3.85 | 1.85 | 0.00 | 0.23 | 0.10 | 1.17 | 0.16 | 0.06 |
| ROTOR | Rotor Squirrel Cage | 143.34 | 33.83 | 0.03 | 2.76 | 1.90 | 45.49 | 1.49 | 0.52 |
| | Rotor Core Laminations | 103.55 | 28.93 | 0.03 | 4.18 | 5.64 | 49.74 | 56.15 | 1.20 |
| | Rotor Shaft | 74.77 | 19.46 | 0.04 | 3.37 | 4.19 | 35.20 | 50.82 | 0.88 |
| | Key Shaft | 0.32 | 0.09 | 0.00 | 0.01 | 0.02 | 0.16 | 0.26 | 0.00 |
| | Fan | 25.47 | 8.07 | 0.00 | 0.76 | 0.31 | 15.30 | 0.29 | 0.13 |
| | Fan Clamps | 0.25 | 0.07 | 0.00 | 0.01 | 0.01 | 0.12 | 0.17 | 0.00 |
| FRAME | Electric Motor Case | 198.49 | 62.99 | 0.04 | 6.00 | 2.64 | 119.72 | 3.82 | 1.02 |
| | Flange Drive-End Shield | 148.00 | 46.76 | 0.03 | 4.52 | 2.05 | 88.74 | 3.56 | 0.77 |
| | Non Drive-End Shield | 101.23 | 32.19 | 0.02 | 3.03 | 1.29 | 61.20 | 1.51 | 0.51 |
| | Terminal Box | 27.80 | 8.80 | 0.01 | 0.85 | 0.40 | 16.75 | 0.76 | 0.15 |
| | Grease Fitting | 0.10 | 0.03 | 0.00 | 0.00 | 0.00 | 0.04 | 0.06 | 0.00 |
| | Grease Fitting Protection | 0.02 | 0.01 | 0.00 | 0.00 | 0.00 | 0.01 | 0.00 | 0.00 |
| | Bearing Drive-End Shield | 10.37 | 3.05 | 0.00 | 0.39 | 0.50 | 4.43 | 6.33 | 0.11 |
| | Bearing Non Drive-End Shield | 10.37 | 3.05 | 0.00 | 0.39 | 0.50 | 4.43 | 6.33 | 0.11 |
| | Fan Cover | 50.14 | 14.20 | 0.02 | 1.89 | 1.71 | 40.23 | 2.51 | 0.53 |
| | Drip Cover | 19.34 | 5.62 | 0.00 | 0.51 | 0.35 | 18.50 | 0.60 | 0.16 |
| MIX | Cables | 13.90 | 5.07 | 0.01 | 1.94 | 36.88 | 225.22 | 61.32 | 0.27 |
| | Miscellaneous (Gaskets, Screws, etc.) | 4.53 | 1.25 | 0.00 | 0.35 | 0.40 | 2.43 | 1.79 | 0.03 |

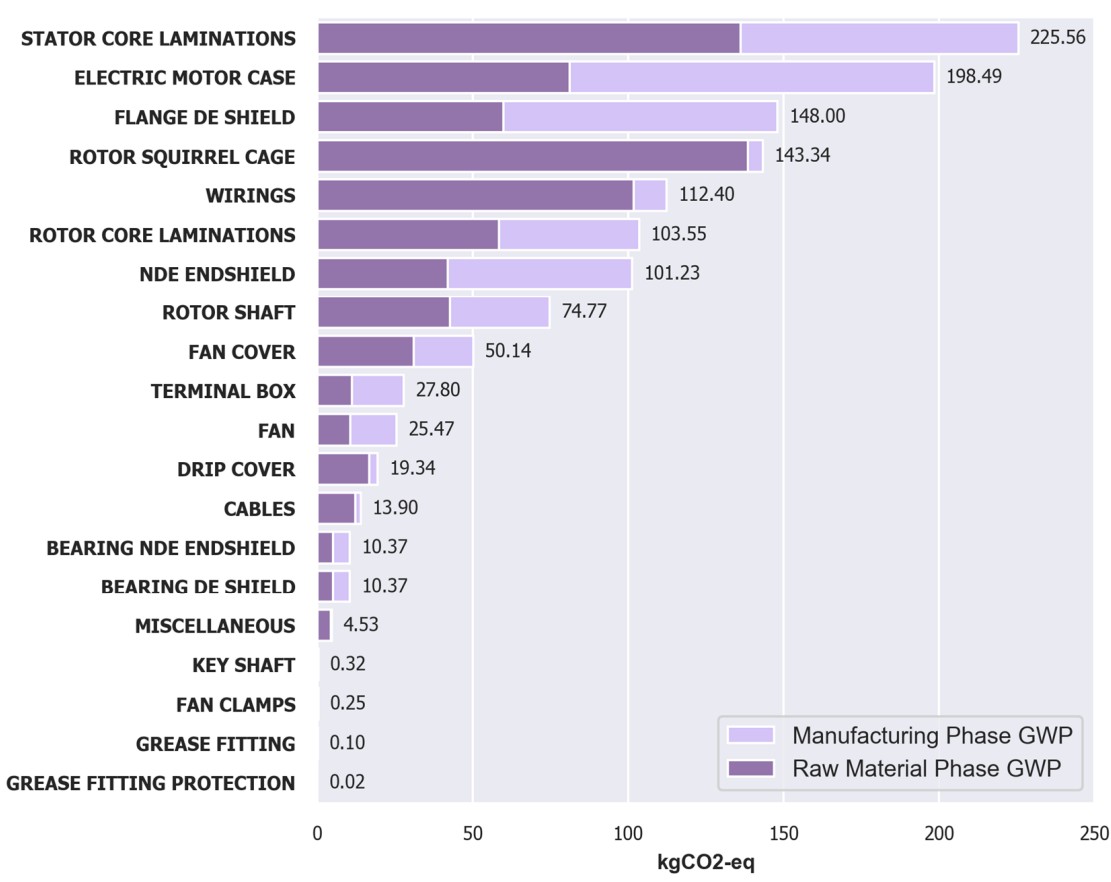

**Figure 6.** GWP outcomes for each AEM component, subdivided between MAT and MAN phases.

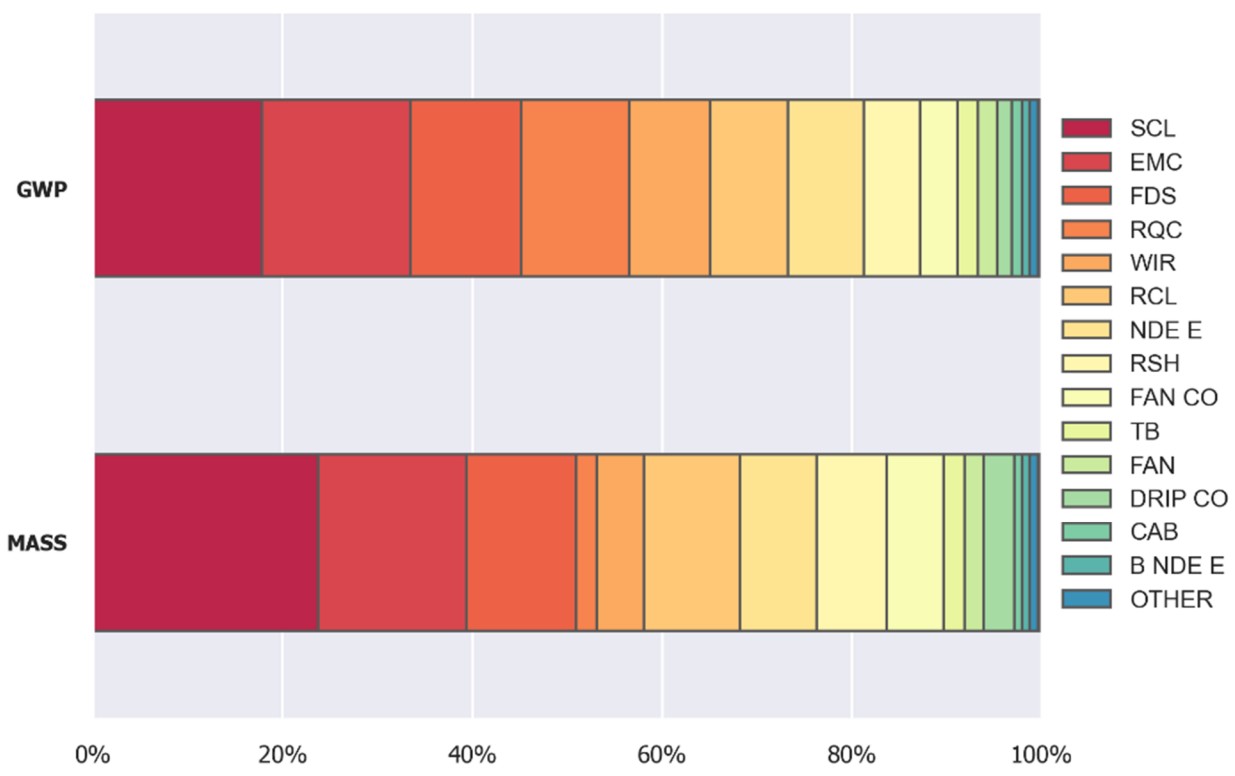

**Figure 7.** Cumulative representation of main AEM components for GWP and mass (OTHER represents the list of the following components: BDS, MISC, KSH, FAN CL, GF, and GFP).

Moreover, the above outcomes suggest a non-proportional relationship between the GWP and the mass of the single components. While one might intuitively expect a direct and extensive correlation between mass and environmental impact, the analysis reveals a weaker relationship since there are processes that do not depend directly on mass but on other characteristic dimensional units (such as length, surface area, worked hours, mass removed, etc.). Several components exhibit lower environmental impacts due to the combination of materials and processes involved: for instance, the windings (WIR) demonstrate a relatively low impact due to the energy-efficient filament-bending process. On the other hand, there are cases where components have higher impacts despite their lower contribution to the overall system mass. For example, the squirrel cage (RQC) shows a high impact due to aluminum extraction and production, despite being a material known for its low density.

In other words, the sole size (or mass) of a component/total motor does not necessarily correlate with the extent of its environmental footprint. This result emphasizes the significance of considering components' shapes, materials used, and manufacturing processes in determining environmental impacts.

Figure 8 depicts the GWP results for the sub-assemblies defined above for the electric motor (refer to Table 3). Similar to the previous figures, this representation utilizes a bar chart format to highlight the impact of each sub-assembly, divided between the raw material (RAW) and manufacturing (MAN) phases, and classified from the most impactful to the least impactful.

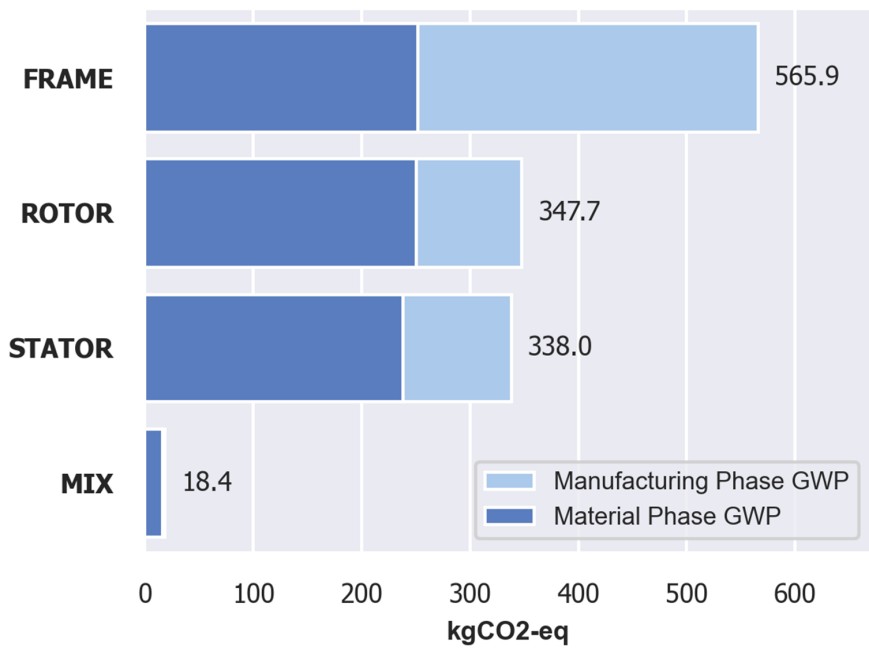

**Figure 8.** GWP outcomes for main AEM sub-assemblies, subdivided between MAT and MAN phases.

Consistent with the findings at the component level, it can be observed that no single phase of the life cycle can be unequivocally deemed more critical. One remarkable finding is the high impact associated with the FRAME sub-assembly, which ranks as the most impactful. This result can be mainly attributed to the sand-casting process, which exhibits a significantly high specific impact. On the other hand, when examining the ROTOR and STATOR sub-assemblies, the RAW phase holds greater importance with regard to percentage contribution, primarily due to the production of metal materials such as steel, aluminum, and copper, which are critical materials used in these sub-assemblies.

Regarding the stator winding component (WIR), remarks about the selected material must be made. In addition to copper, manufacturers have proposed aluminum stator windings to reduce the cost and weight of electric motors with the same specifications and

efficiency [34,35]. Aluminum's resistivity is more than 50% higher than copper's resistivity even though its density is around 30% lower than that of copper. Due to this aspect, a greater aluminum winding volume is required to maintain the same characteristics. This circumstance led to different lightweight design solutions for a larger outer stator diameter or smaller inner rotor diameter [36]. From a sustainability point of view, the specific impacts at the material stage of aluminum and copper differ, and these differences depend on the impact category analyzed. For example, looking at GWP (kg $CO_{2eq}$/kg material), the specific impact of aluminum is three times higher than the GWP-specific impact of copper. In contrast, for MDP (kg $CO_{2eq}$/kg material), the specific impact of copper is more than 150 times higher than that of aluminum. A further evaluation of all the ReCiPe impact categories is needed to determine whether or not aluminum winding could be a sustainable solution due to the difference in density and resistivity between the materials considered.

Relative to efficiency, some considerations regarding the scalability of the environmental impacts are necessary. Moving from the evaluated efficiency class (High-Efficiency IE2) to Premium Efficiency (IE3) and then to Super-Premium Efficiency (IE4), the efficiency improvement is about 1–3% per class for induction electric motors with powers greater than 10 kW [37]. In order to achieve these improvements, the manufacturers apply different solutions that vary the material composition and mass breakdown, as proposed in the section above [38]:

- Increasing the copper content and stator conductor sectional area, resulting in a reduction in stator resistance.
- Increasing the copper content and rotor dimension, leading to a reduction in rotor resistance.
- Increasing the silicon content and reducing the size of the core lamination, and thus decreasing the magnetizing losses.

All of these methods led to a clear increase in the environmental burden in both the RAW and MAN phases, including more impactful materials, like silicon, and more production steps, like core lamination $CO_2$ lasering. In addition, the electric motor efficiency can also be enhanced by optimizing sensors or other components, like inverters or cooling systems, all outside of the system boundary analysis defined for this paper.

Assessing the different commercial solutions available in the market, each manufacturer proposes its own solution to achieve the efficiency standard (IE2, IE2, and IE3) and insulation (A, B, F, and H) classes. With regard to the first aspect, Bortoni et al., 2019 [38], proposed a model to estimate the rotor, stator, and magnetized losses for different commercial induction electric motors with the same efficiency class. The authors linked those losses variation to different design solutions or material compositions. Following these considerations, little variations in all of the environmental impacts are expected due to the appreciable structural differences between commercial electric motors. Even for insulation, significant differences in the environmental results are assumed, especially when impactful materials are used extensively, like epoxy resins for vacuum encapsulation technology [39]. In order to define a comprehensive method for evaluating the sustainability of the efficiency class, insulation level, and commercial models, it is necessary to further analyze the leading solutions proposed in the literature.

It appears clear that to quantify the variability of the environmental spectrum in each efficiency and insulation class and between different commercial models (same power), further analysis is needed, rigorously considering the mass breakdown and technical specifications of different solutions in the market.

In Figure 9, eight of the indicators defined by the ReCiPe methodology are presented; the others can be found in the Supplementary Materials. These indicators provide a comprehensive framework for assessing the various environmental impacts associated with the electric motor. Concerning the FDP and WDP indicators, the results exhibit a consistent trend with the findings observed for the GWP. This alignment suggests that the above categories accurately capture and reflect the electric motor's environmental implications, particularly regarding climate change impacts.

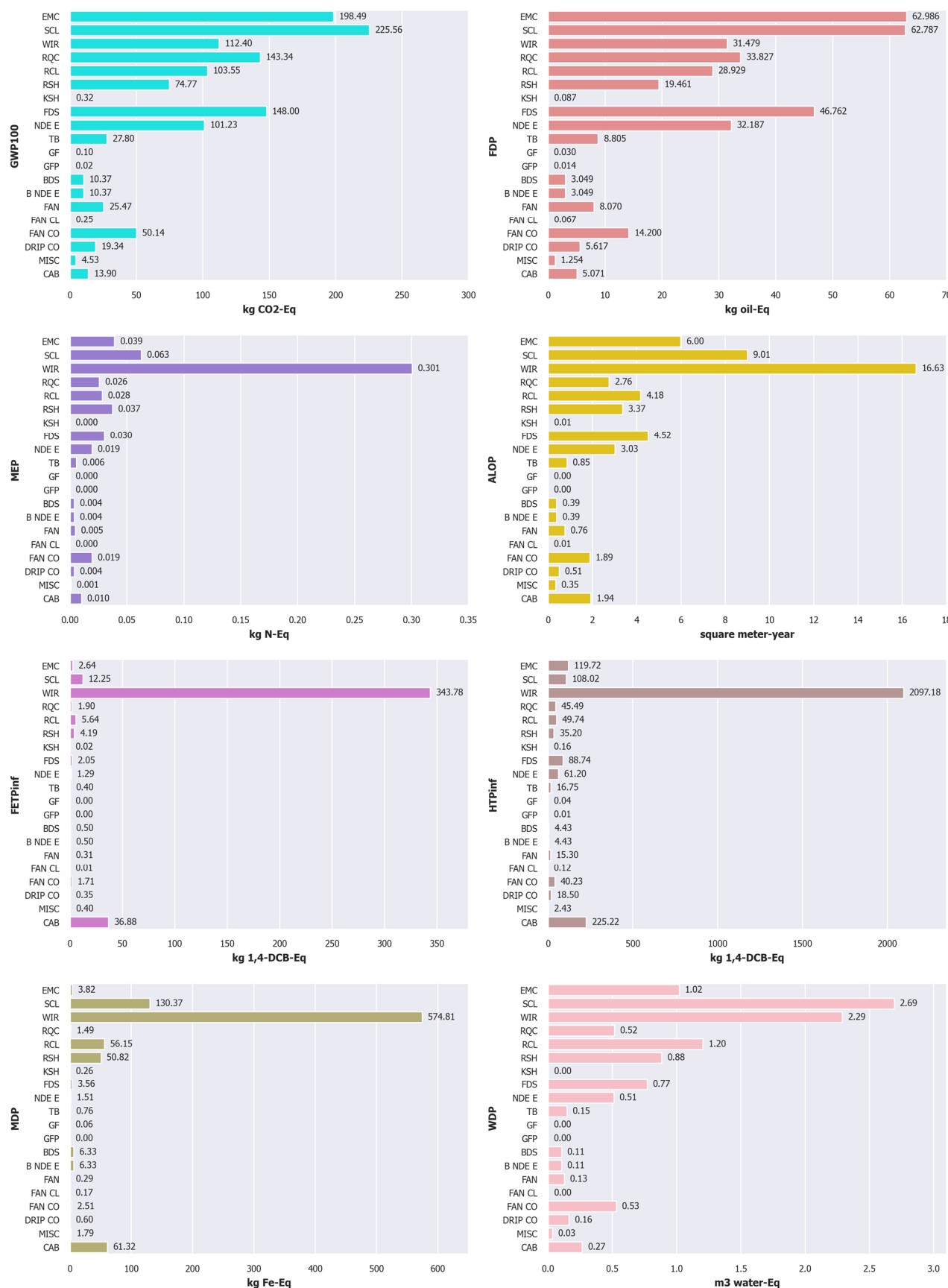

**Figure 9.** ReCiPe impact category outcomes of AEM components.

However, when examining the other indicators, it can be observed that there is no strong correspondence with what was observed for the environmental footprint. Different indicators shed light on specific environmental aspects, offering insights beyond greenhouse gas emissions alone, thus highlighting the importance of considering multiple impact categories to gain a comprehensive understanding of the overall environmental performance of the electric motor.

For instance, the WIR component (i.e., stator windings) stands out with the highest values across many indicators (such as $FETP_{inf}$, $HTP_{inf}$, MDP or MEP), indicating its significant environmental impact. While the impact may not be as high as that of GWP, it remains substantial. This effect can be attributed to the production of copper, which affects multiple indicators due to its resource- and emission-intensive nature, and the use of epoxy resins in stator insulation, which has an enormous impact in categories such as MEP, FEP and NLTP (see the Supplementary Materials). Switching to the least impactful insulation materials, like heat-resistant fibers or biodegradable thermoplastics, could be a game changer with regard to the environmental burden of induction electric motors [40]. Similarly, the CAB component (i.e., motor cables) exhibits a similar pattern to WIR since the cabling is mainly made of copper. Consequently, the environmental impacts associated with the production of copper are evident in both the WIR and CAB components.

## 4. Conclusions

This life cycle assessment study provides guidelines for designers regarding the environmental sustainability of the analyzed product: an asynchronous electric motor for stationary applications.

Based on a state-of-the-art review, this work intends to address the following significant aspects: the implementation of a top-down scientific approach (i.e., the modeling tree inventory data implementation), aimed at quantitative modeling to overcome the scarcity of sensitive data for industrial applications, using a specific CAD reconstruction. Therefore, a cradle-to-gate analysis was conducted, with a particular emphasis on detailed modeling of the manufacturing stage, revealing insights that may go unnoticed in a cradle-to-grave evaluation.

The above results delve into the details of each component and specific sub-assemblies, covering several fundamental aspects.

At the level of the individual AEM parts, identifying components with a significant impact despite their comparatively lower mass can guide decision-making processes toward optimizing these specific components for environmental performance. This highlights the need to consider factors beyond mass when assessing the overall sustainability of the system. By recognizing this non-proportional relationship between the GWP and mass, it becomes clear that focusing solely on mass reduction may not necessarily lead to the most significant environmental improvements. A holistic approach that considers material choices, production processes, and the functional requirements of each component is essential for achieving sustainable outcomes. In conclusion, the observation that the impact of the first six components (electric motor case (EMC), stator core laminations (SLCs), wiring (WIR), rotor squirrel cage (RQC), rotor core laminations (RCLs), and rotor shaft (RSH)) represents a substantial percentage of the total system mass and environmental impact challenges the assumption of a non-linear proportionality between mass and environmental impact. This finding underscores the importance of a comprehensive assessment considering various environmental performance factors. By considering these insights, both designers and engineers can make decisions to optimize the environmental sustainability of the components and the overall system.

Concerning the sub-assembly perspective, the analysis presented in Figure 6 provides valuable insights into the GWP impacts associated with sub-assemblies of the electric motor. Similar to the observations made at the component level, there is no singularly dominant life cycle phase.

The effects on the sustainability of the efficiency class, insulation level and manufacturer design choices, like aluminum winding, are also evaluated, highlighting the principal hotspots for each classification or solution and defining the guidelines for further detailed analysis.

The findings emphasize the importance of considering both the raw material (RAW) and manufacturing (MAN) phases and highlight specific guidelines for environmental improvement. Understanding the environmental implications of different sub-assemblies enables informed decision making and the development of sustainable strategies that target specific processes and materials. By addressing these findings, it becomes possible to enhance the overall environmental performance of electric motors and contribute to the broader goal of achieving sustainability in the industrial sector. The FRAME sub-assembly's high manufacturing (MAN) phase impact suggests that optimizing or re-evaluating the sand-casting process could mitigate its environmental footprint. Consequently, strategies to reduce the environmental impact of other sub-assemblies (i.e., ROTOR and STATOR) should focus on material sourcing, efficient material use, and recycling initiatives. These findings emphasize the importance of considering the production and use of copper and insulation materials in the electric motor's life cycle analysis. Efforts to optimize copper usage, explore alternative materials, and enhance recycling practices can help mitigate the environmental impacts associated with these components.

Moreover, the analysis of the 18 indicators defined by the ReCiPe methodology provides a comprehensive understanding of the environmental performance of the electric motor. While there is coherence between the results of the FDP and GWP, indicating the importance of climate change impacts, other indicators offer additional insights into diverse environmental aspects. For instance, the high values associated with the WIR and CAB components, driven by the production of copper and epoxy resin, highlight the need for targeted actions to address their environmental footprints. Integrating this knowledge into decision-making processes makes it possible to develop strategies that consider multiple environmental impact categories and work towards the sustainable development of electric motors (such as material sourcing, efficient material use, and circular economy approaches).

Finally, this study reaffirms the need to consider the entire life cycle of the motor when evaluating environmental impacts and underscores the importance of a holistic approach.

**Supplementary Materials:** The following supporting information can be downloaded at https://www.mdpi.com/article/10.3390/machines11080810/s1: Table S1: Brief description of ReCiPe midpoint indicators; Table S2: Activities directly associated with Ecoinvent Database; Table S3: Activities (user-defined processes) modeled with Ecoinvent Database; Figure S1 and Table S4: Ten out of all eighteen ReCiPe impact category outcomes for each AEM's components.

**Author Contributions:** Conceptualization, A.A. and A.G.; methodology, A.A., A.G. and E.I.; software, A.A.; validation, A.A., A.G. and E.I.; formal analysis, A.A., A.G. and E.I.; investigation, A.A., A.G. and E.I.; resources, A.A., A.G. and E.I.; data curation, A.A., A.G. and E.I.; writing—original draft preparation, A.A., A.G. and E.I.; writing—review and editing, A.A., A.G., E.I. and M.D.; visualization, E.I.; supervision, M.D.; project administration, M.D. All authors have read and agreed to the published version of the manuscript.

**Funding:** This research received no external funding.

**Data Availability Statement:** Shareable data are all reported in the paper and related Supplementary Materials. Further details cannot be provided due to a pending NDA agreement.

**Acknowledgments:** The authors wish to thank Lorenzo Berzi (Department of Industrial Engineering, University of Florence) for the technical support.

**Conflicts of Interest:** The authors declare no conflict of interest.

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
