# Peer review of "A Scientific Approach for Environmental Analysis: An Asynchronous Electric Motor Case Study for Stand-By Applications"

_machines, doi:10.3390/machines11080810_

Round 1

Reviewer 1 Report

Typo at line 121, Table S1, S2, etc.

The use of acronyms make the paper difficult to follow

Although pointing to the standards and methods, the paper should contain further details on how the presented indicators have been obtained

As the motor is modeled in detail, some physical parameters should be presented, to confirm the results and conclusions (for example to detail the motor components individual weight, and further, to detail how the GWP outcomes in Fig 6 have been obtained)

At line 290 Figure XX remained unnumbered

Author Response

Dear Reviewer,

Please find attached the reviewed paper according to your comments.

The authors are looking forward to your kind feedback.

Best regards,

The authors

Reviewer 2 Report

Comment to the authors

1.- Complete name plate of the motor should be included in the manuscript.

2.- The breakdown of the different component of the motor is valid for any manufacturer / efficiency class?

3.- Stator

In the Table 2, the stator comprises two components, core laminations and wirings.

What about the insulation? This is a key part of the machine.

In my opinion this is an important issue from the point of view of the paper.

Author Response

(The authors gave the same response as above.)

Reviewer 3 Report

Dear authors, the article is interesting and easy to read. I would suggest to add some considerations relating to the efficiency classes of electric motors, in particular what impact do you expect in the transition from class IE2 to IE3 to IE premium.
Furthermore, some considerations concerning the use of aluminum windings would be interesting.

Author Response

(The authors gave the same response as above.)

Reviewer 4 Report

It is my opinion that you should provide adequate documentation confirming the Recipe 1.13 methodology vs other Recipe versions and vs other approaches (LIME, EPS, ...).

Moreover, you should analyze how you graded your case study components in each impact category.

Please, identify the 6 components in line 284 and the Figure XX in line 290.

Author Response

(The authors gave the same response as above.)

Round 2

Reviewer 4 Report

Improvements are made to a satisfactory level.